# M2 Monocyte Polarization in Dialyzed Patients Is Associated with Increased Levels of M-CSF and Myeloperoxidase-Associated Oxidative Stress: Preliminary Results

**DOI:** 10.3390/biomedicines9010084

**Published:** 2021-01-16

**Authors:** Valérie Pireaux, Cédric Delporte, Alexandre Rousseau, Jean-Marc Desmet, Pierre Van Antwerpen, Martine Raes, Karim Zouaoui Boudjeltia

**Affiliations:** 1URBC-Narilis, University of Namur, 61 rue de Bruxelles, 5000 Namur, Belgium; valerie.pireaux@gmail.com (V.P.); martine.raes@unamur.be (M.R.); 2Laboratory of Pharmaceutical Chemistry and Analytical Platform, Faculty of Pharmacy, Université libre de Bruxelles (Campus de la Plaine) CP205/05, Boulevard du Triomphe, 1050 Brussels, Belgium; Cedric.Delporte@ulb.be (C.D.); Pierre.Van.Antwerpen@ulb.be (P.V.A.); 3Laboratory of Experimental Medicine (ULB 222 Unit), CHU-Charleroi, ISPPC Hôpital Vésale, Université libre de Bruxelles, 6110 Montigny-Le-Tilleul, Belgium; alexandre.rousseau@chu-charleroi.be; 4Nephrology-Hemodialysis Unit, CHU-Charleroi, ISPPC Hôpital Vésale, 6110 Montigny-Le-Tilleul, Belgium; jeanmarc.desmet@chu-charleroi.be

**Keywords:** dialysis, inflammation, monocytes, myeloperoxidase, oxidation, polarization

## Abstract

Cardiovascular diseases represent a major issue in terms of morbidity and mortality for dialysis patients. This morbidity is due to the accelerated atherosclerosis observed in these patients. Atherosclerosis is a chronic inflammatory disease characterized by key players such as monocytes, macrophages, or oxidized LDLs. Monocytes-macrophages are classified into subsets of polarized cells, with M1 and M2 macrophages considered, respectively, as pro- and anti-inflammatory. (1) Methods: The monocyte subsets and phenotypes were analyzed by flow cytometry. These data were completed by the quantification of plasma M-CSF, IL-8, CRP, Mox-LDLs, Apo-B, Apo-AI, chloro-tyrosine, and homocitrulline concentrations. The statistical differences and associations between two continuous variables were assessed using the Mann–Whitney U test and Spearman’s correlation coefficient, respectively. (2) Results: Hemodialyzed patients showed a significant increase in their concentrations of CRP, M-CSF, and IL-8 (inflammation biomarkers), as well as chloro-tyrosine and homocitrulline (myeloperoxidase-associated oxidative stress biomarkers). Moreover, we observed a higher percentage of M2 monocytes in the plasma of hemodialysis patients as compared to the controls. (3) Conclusions: Our data suggest that oxidative stress and an inflammatory environment, which is amplified in hemodialysis patients, seems to favor an increase in the concentration of circulating M-CSF, therefore leading to an increase in M2 polarization among circulating monocytes.

## 1. Introduction

Cardiovascular diseases (CVD) account for approximately 30% to 40% of deaths among dialysis patients ([1], updated 19 September 2016). They represent a major issue in terms of morbidity and mortality for the latter, and specifically in the case of hemodialyzed (HD) patients showing a 5-year survival rate on stable HD (for a review, see [2]). The morbidity associated with CVD is due to accelerated atherosclerosis, which has been shown to be present in the carotid arteries of HD patients as opposed to aged-matched healthy controls, with a higher thickness of the media/intima as well as with arterial stiffness and calcification within the plaques [3,4,5,6].

Atherosclerosis is a chronic inflammatory disease affecting mainly medium- and large-sized arteries. Indeed, injuries or local blood flow perturbations lead to increased permeability of the endothelial layer, favoring lipoprotein infiltration in the intima where they are oxidized and become atherogenic (for reviews, see [7,8,9]). This oxidation activates endothelial cells, enhancing chemokine and cytokine secretion and finally inducing the recruitment of monocytes that will differentiate into macrophages in the intima [8,9].

Monocytes-macrophages are important key players in the initiation and evolution of atherosclerotic lesions. They display a high plasticity and heterogeneity and are activated by different signals varying according to the cellular environment, modulating their phenotypes as an adaptive response. The classification of monocytes and macrophages is based on the nomenclature of Th1/Th2 immune cells. In vitro, the M1 phenotype corresponding to “classically-activated macrophages” is obtained when they are exposed to IFNγ (Interferon gamma) combined with a TNFα (Tumor Necrosis Factor alpha) inducer such as LPS (Lipipolysaccharides) or to GM-CSF (Granulocytes macrophage-colony stimulating factor) [10,11,12,13]. M1 macrophages display pro-inflammatory properties and are mostly involved in acute host defense towards intracellular microorganisms, tumor resistance, and tissue destruction thanks to their capacity to produce bursts of reactive oxygen and nitrogen species and secrete pro-inflammatory cytokines (e.g., IL-6 and IL-8) [11,14]. On the other hand, anti-inflammatory M2 macrophages resolve inflammation by the production of anti-inflammatory mediators [14,15]. They take part in matrix remodeling, angiogenesis, tissue repair and healing, but can also be immunosuppressive, promoting tumor development [16,17]. This phenotype can be induced by macrophage-colony stimulating factor (M-CSF), by IL-4 and IL-13 (M2a), by immune complexes, by TLR (Toll like receptor) agonists or IL-1R (M2b), or by IL-10, as well as by glucocorticoid or secosteroid hormones (M2c) [10,11,12,18]. M2a macrophages are characterized by the expression of markers such as CD206, CD163, and dectin-1; M2b by the expression of IL-10, MHC-II, and CD206; and M2c by the expression of CD163, CD206, and Tie-2 [13]. In this study, we decided to keep the M1/M2 classification, even though we are aware that it is mainly based on in vitro studies that do not adequately mirror the complexity of the cell environment in vivo.

In the literature, the polarization of macrophages has been much more documented over the last few years in macrophages compared to monocytes. However, the term “polarization” has also been suggested to be of interest for circulating monocytes [19].

The cause of accelerated atherosclerosis in dialysis patients is not known yet, even though several major risk factors are well represented, such as high blood pressure, higher concentrations of plasma lipids, inflammation (e.g., elevated homocysteine, CRP (C-reactive protein), or fibrinogen levels), oxidative stress, hyperinsulinemia, mineral metabolism disorders, or anemia (for reviews, see [20,21]).

In this study, we investigated whether the polarization of circulating monocytes would be different in hemodialysis patients versus healthy volunteers and whether it could be correlated with the inflammatory status, as assessed by quantifying inflammatory proteins (IL-8, CRP, and M-CSF) in the plasma. Oxidative stress was also monitored by assessing homocitrulline (Hcit). Homocitrulline is a product of protein carbamylation, generated non-enzymatically from lysine residues by the action of cyanate. Cyanate is itself derived from the spontaneous decomposition of urea or via the oxidation of thiocyanate by myeloperoxidase in the presence of H_2_O_2_ [22]. Finally, to gain insight into the lipoprotein status, Apo-AI, Apo-B, and also myeloperoxidase-oxidized LDLs (Mox-LDLs) were quantified in dialysis patients versus healthy volunteers.

## 2. Methods

### 2.1. Study Participants

A total of 27 patients undergoing hemodialysis and 23 healthy volunteers were studied (see Appendix A in Appendix A). They were recruited at the nephrology unit of the CHU-Charleroi, ISPPC Hôpital Vésale in Belgium. 

Written informed consent was obtained from each patient and healthy donor included in the study. The protocol of the study was in conformity with the ethical guidelines of the Helsinki Declaration of 1975 (revised in 2000) and was approved by the institution’s ethics committee (N°: OM008; P15/50_18/11, Eudract: B325201526257). 

### 2.2. Flow Cytometry

A total of 100 µL of total blood from patients was incubated for 15 min at room temperature with PE mouse anti-human CD14 and V500 mouse anti-human CD16 antibodies (Becton Dickinson, Franklin Lakes, NJ, USA) as well as with mouse anti-human CD86-FITC and anti-human CCR2-APC monoclonal antibodies (Miltenyi Biotec, Bergisch Gladbach, Germany) for determining the M1 polarization, or with mouse anti-human CD206-FITC, anti-human CXCR3-APC, and anti-human CD163-VioBlue monoclonal antibodies (Miltenyi Biotec, Bergisch Gladbach, Germany) for the M2 polarization. Red blood cells were then eliminated by adding BD FACS Lysing Solution (dilution: 1/20) (Becton Dickinson, Franklin Lakes, NJ, USA) to the total blood and the remaining cells were washed twice with 1 mL of running buffer. Cells were finally resuspended in 300 µL of running buffer for analysis. The matching isotype controls were used for each antibody in order to define the threshold. The analysis was performed using the MACSQuant Analyzer 10 (Miltenyi Biotec, Bergisch Gladbach, Germany), applying a gating strategy based on the SSC vs. PE gate (CD14), selecting the monocyte population. Classical monocytes were defined based on a high expression of CD14 and a low expression of CD16 (CD14+CD16−). Intermediate monocytes were defined based on a high expression of CD14 and CD16 (CD14+CD16+), while non-classical monocytes were characterized by a low expression of CD14 and a high expression of CD16 (CD14−CD16+).

### 2.3. Measurements of Myeloperoxidase-Modified LDLs, M-CSF and IL-8 (ELISA Assays)

The antibodies used for the measurement of myeloperoxidase-modified LDLs (Mox-LDLs) have previously been fully characterized [23]. They react with the ApoB-100 protein moiety and provide positive signals in human atherosclerotic lesions [23].

Human serum IL-8 and M-CSF ELISA assays were performed following the manufacturer’s instructions (IL-8: BD Biosciences, Franklin Lakes, NJ, USA; M-CSF: R & D Systems, Minneapolis, MN, USA).

### 2.4. Quantitative Analysis of CRP, Apo-AI and Apo-B

Serum parameters such as hs-CRP, Apo-AI, and Apo-B were evaluated by antibody-binding and turbidity measurement on the SYNCHRON LX^®^. 

### 2.5. Homocitrulline, Lysine (Lys), Chloro-Tyrosine (Cl-Tyr), and Tyrosine (Tyr) Quantification by LC-MS/MS

Protein-bound homocitrulline and chloro-tyrosine were monitored in plasma after acid hydrolysis as previously described [24]. Briefly, plasma (20 µL) was placed into the vial and 200 µL of acid mixture (6 M HCl supplemented with 0.05% (*m*/*v*) phenol) and internal standards were added. Hydrolysis was carried out for 35 min at 110 °C. 13C9-Tyr and 13C15N-Lys were used as internal standards. Samples were evaporated to dryness under nitrogen flow, labeled with a butanolic HCl solution, dried under nitrogen flow, and finally dissolved in 1.0 mL formic acid 0.1% in water before injection into the LC-MS system. The LC system was a 1290 Infinity series UHPLC system (Agilent Technologies, Palo Alto, CA, USA). Amino acid residues were resolved on a Poroshell 120 EC-C18 column (2.1 × 100 mm, 2.7 µm) (Agilent Technologies, Palo Alto, CA, USA) using a gradient of 0.2% formic acid and methanol. Amino acid residues were quantified by tandem MS on a 6490 series ESI-triple quadrupole mass spectrometer using a JetStream source (Agilent Technologies, Palo Alto, CA, USA). Data were acquired using the MassHunter Acquisition^®^ software and analyzed by the MassHunter Quantitative Analysis^®^ software (Version B.07, Agilent Technologies, Santa Clara, United States). Data were expressed as the ratio of homocitrulline to lysine and ratio of chloro-tyrosine to tyrosine.

### 2.6. Statistical Analysis

Statistical analyses were performed using the SigmaPlot (version 12.2) software (Systat Software, San Jose, CA, USA).

The Shapiro–Wilk test was used to assess normal distribution. Continuous non-normally distributed data were expressed as a median and interquartile range, median (25–75%). They were analyzed by the Mann–Whitney U test. A correlation was checked using Spearman’s *r* coefficient. Differences were considered significant when *p* < 0.05.

## 3. Results

### 3.1. Volunteers and Patients

Healthy volunteers (23) did not present any cardiovascular risk factor (menopausal status, diabetes, hypertension, smoking habit, history of coronary or stroke event). 

The main characteristics of the dialyzed patients (27) are shown in the Appendix A. Hemodialyzed patients’ parameters—creatinine: 6.5 (4.7–10.05) mg/dL; GFR (MDRD): 7.1 (5.1–10.6) mL/min; Weight Inter Dial: 2000 (1500–3500) g.

We observed a significant difference in age between healthy subjects (55 (52–58) years) and dialyzed patients (71 (63–83) years); *p* < 0.001.

### 3.2. Circulating Anti-Inflammatory and Immunosuppressive Monocytes Are Increased in HD Patients

At first, the monocyte phenotype and polarization were investigated in hemodialyzed patients (HD) versus non-HD donors. Based on the CD14 and CD16 expression, we observed a significant decrease in the percentage of classical CD14+CD16− monocytes in HD patients compared to healthy subjects (controls) (*p* < 0.05), while the percentages of intermediate CD14+CD16+ and non-classical CD14−CD16+ monocytes was similar in the two groups (Figure 1A).

Then, assessing the expression of M1 (CD86 and CCR2) and M2 (CD206, CXCR3, and CD163) protein markers, we observed that, in comparison to the healthy subjects, HD patients showed an increased percentage of anti-inflammatory and immunosuppressive CD206+CXCR3+CD163+ M2 monocytes (*p* < 0.01), while the pro-inflammatory CD86+CCR2+ M1 monocyte population did not seem to be modified (*p* > 0.05) (Figure 1B,C). While analyzing the M1 or M2 predominance in the different monocyte subpopulations, we observed an increase in the M2 CD14+CD16+ monocyte subpopulation (*p* < 0.001), as well in the M2 CD14−CD16+ monocytes (*p* < 0.05), in HD patients as opposed to the controls. There was no significant difference in the M1 CD86+CCR2+ monocyte subpopulations (Figure 2A,B).

### 3.3. Biomarkers of Inflammation, Including M-CSF, Are Increased in HD Patients

Because M-CSF is described as an inflammation biomarker, known to favor macrophage polarization, we decided to measure the blood concentration of M-CSF but also of other recognized inflammation markers such as IL-8 as well as CRP, shown to induce M-CSF secretion by endothelial cells (Figure 3).

As a result, we observed that the concentration of M-CSF was significantly higher in the blood of HD patients as compared to that of healthy individuals (*p* < 0.001) (Figure 3A). These results are in agreement with the data of Nitta et al. (2001) and Nishida et al. (2016) [25,26]. The plasma CRP as well as IL-8 concentration was also increased in HD patients in comparison with the controls (respectively *p* < 0.01 and *p* < 0.001) (Figure 3B,C).

### 3.4. HD Patients Undergo Myeloperoxidase-Dependent Oxidative Stress

Besides inflammatory proteins and M-CSF, we also measured the plasma concentration of myeloperoxidase (MPO)-oxidized LDLs (Mox-LDLs) and chloro-tyrosine (Cl-Tyr), a product of LDL oxidation by MPO (Figure 4). The abundances of apolipoprotein-AI (a constituent of High-Density Lipoproteins) and Apo-B (a constituent of Low-Density Lipoproteins) were also assessed (Figure 4).

Interestingly, an increase in the plasma concentration of Mox-LDLs (ratio of Mox-LDLs/Apo-B) was observed in hemodialysis patients, even though this trend was not significant (*p* = 0.11) (Figure 4A). The concentration of chloro-tyrosine (Cl-Tyr) also increased in the plasma of HD patients, as compared to healthy controls (*p* < 0.001) (Figure 4D). This increase positively correlated with an increase in the percentage of M2 monocytes (r = 0.43; *p* = 0.001) in HD patients and healthy controls (Figure 5). We also observed that M-CSF increased concentration is positively correlated with the increased percentage of M2 monocytes in HD patients (r = 0.48; *p* < 0.0004) (Figure 5). In addition, a significant decrease in the Apo-AI concentration, the major apolipoprotein of high-density lipoprotein, was observed in the blood of HD patients as opposed to the controls (*p* = 0.002) (Figure 4C).

Finally, the concentration of homocitrulline was quantified. Homocitrulline is a product of protein carbamylation that accumulates after the oxidation of thiocyanate by myeloperoxidase in the presence of H_2_O_2_ [22]. An increase in protein carbamylation has been observed in inflammatory diseases such as atherosclerosis [22]. The plasma concentration of homocitrulline (Hcit) increased in HD patients, as compared to the controls (Figure 4E).

## 4. Discussion

Our data suggest that oxidative stress and an inflammatory environment, amplified in hemodialysis (HD) patients, seems to favor an increase in the concentration of circulating M-CSF, associated with an increase in M2 polarization among monocytes in the bloodstream.

Accelerated atherosclerosis is a major problem in terms of morbidity for HD patients. Monocytes and macrophages are key cells in the pathophysiology of atherosclerosis. Indeed, they contribute to lesion development after the infiltration of the monocytes into the intima and their differentiation into macrophages. Besides producing inflammatory mediators contributing to the evolution of lesions, macrophages also accumulate modified LDLs and become foam cells, a hallmark of atherosclerotic lesions [9].

In this study, we decided to investigate monocytes in the blood of HD patients, taking into account their polarization. Unexpectedly, we observed an increase in the percentage of circulating M2 monocytes.

Therefore, we wondered why the percentage of circulating M2 monocytes increased in hemodialysis patients. M-CSF, a hematopoietic growth factor, considered as a potent cytokine, is a key player in monocyte differentiation into macrophages, regulating monocyte and macrophage survival, proliferation, and activation [10,27]. In agreement with other studies [25,28], we showed that the M-CSF concentration was higher in the plasma of HD patients in comparison to healthy individuals.

The M-CSF level in the circulation increases in various pathologies such as cancer, infections, and chronic inflammatory diseases [29,30,31]. It is produced by arterial wall cells such as endothelial cells and fibroblasts, as well as by macrophages in atherosclerotic lesions [32,33,34]. Several studies in mice and humans have linked M-CSF to chronic inflammatory diseases such as atherosclerosis and concluded that M-CSF would have a pro-atherogenic role [35,36]. Indeed, M-CSF has been detected within atherosclerotic lesions, where it is overexpressed as compared to healthy artery tissues [33]. Furthermore, Smith et al. showed that the size of lesions of hyperlipidemic *ApoE*^−/−^ mice, deficient in M-CSF, is reduced [37].

Devaraj et al. showed that CRP, a pro-inflammatory protein, induces the production of M-CSF by endothelial cells and macrophages [38]. We showed here that CRP concentration indeed increased in the plasma of HD patients.

We also quantified the level of plasma IL-8 to further confirm the pro-inflammatory status: the plasma concentration of this protein was also increased in HD patients in comparison with healthy volunteers.

Several teams showed that oxidized LDLs induce IL-8 secretion in cultured endothelial cells [39] and vascular smooth muscle cells [40]. Others recently identified CRP in the supernatants of human aortic endothelial cells stimulated with Ox-LDLs [41]. Besides IL-8, elevated levels of CRP and oxidized LDLs were found to positively correlate with cardiovascular diseases (for a review, see [42]). Mox-LDLs were also shown to trigger IL-8 secretion in endothelial cells [43]. Mox-LDLs are the result of LDL oxidation by myeloperoxidase, a cationic enzyme considered as one of the physiologically relevant oxidative systems of LDLs. In this work, we observed that the serum Mox-LDLs raised in HD patients at the limit of significance while chloro-tyrosines detected on plasmatic proteins significantly increased. These modifications due to MPO activity are also detected in atherosclerotic lesions [23,44]. We also observed a higher level of plasma homocitrulline in HD patients in comparison to the controls. As already mentioned, homocitrulline is a product of protein carbamylation, considered, at least in part, as a biomarker of MPO activity in the presence of H_2_O_2_ [22]. An increase in protein carbamylation has been observed in inflammatory diseases such as atherosclerosis (for a review, see [22]). Carbamylated proteins are indeed important in the atherosclerosis process, as it has been shown that carbamylated LDLs induce monocyte adhesion to vascular endothelial cells [45]. Moreover, carbamylated HDLs have been shown to contribute to foam cell formation [46]. Patients suffering from renal failure with uremia also displayed an increase in protein carbamylation, along with an increase in homocitrulline concentration [47]. This is consistent with our data, showing an increased concentration of plasma homocitrulline in HD patients. Taken all together, our data suggest MPO-dependent oxidative stress in HD patients.

Therefore, it is possible that Mox-LDLs and chloro-tyrosine on plasma proteins, more abundant in HD patients, induce the production of IL-8 and CRP, leading to their increase in the plasma and then to an increase in the plasma M-CSF concentration.

Pireaux et al. showed in vitro that Mox-LDLs enhance an M2 phenotype in murine macrophages [48]. Thus, the increased concentrations of Mox-LDLs and chloro-tyrosine on plasma proteins and M-CSF could explain the increased percentage of M2 monocytes observed in HD patients in our study. Further analyses are needed in order to validate this hypothesis.

In the context of renal pathology, studies have shown that the percentage of M2 monocytes is also modified in the renal stroma of patients. However, the recruitment and the role of CD163+ M2 macrophages in the dysfunctional kidney is not understood yet. There is evidence that they could be involved in the disease progression, with studies showing an association between increased CD163+ M2 macrophage infiltration and decreased renal function. Other studies have shown that CD163+ M2 macrophages are associated with renal dysfunction as it progresses from the acute inflammatory to chronic fibrotic phase as the M2/M1 macrophages ratio increased [49,50]. However, Lu et al. (2013) proposed a role in tissue repair, the restoration of tissue integrity, and the improvement of renal function by showing that CD163+ M2a and M2c macrophages were protective against renal inflammation and renal injury in chronic kidney disease [51].

Thus, it remains unclear whether CD163+ M2 macrophages are primary actors of disease progression or are recruited in order to limit inflammation induced tissue damage as much as possible.

Considering that HD patients suffer from accelerated atherosclerosis, we were surprised to observe an increase in the circulating M2 monocytes. Indeed, M2 polarization among macrophages is usually considered as protective in atherosclerosis. M2 macrophages have been predominantly detected in early lesions of *ApoE^−/−^* mouse lesions [52], but also in regressing plaques [53] or the human perivascular adventitial tissue [54]. But M2 monocytes-macrophages are heterogeneous and can be considered as anti-or pro-atherogenic, depending on the stimuli and on the proteins they express. The increase in M2 monocytes in HD patients, characterized by accelerated atherosclerosis, could then be explained by two hypotheses. First, the M2 polarization is pro-atherogenic, as has been hypothesized for IL-4-induced M2 macrophages [55]. Indeed it has been shown that IL-4 induces the expression of the scavenger receptor CD36 in macrophages, promoting oxidized LDL uptake. IL-4 also upregulates the expression of matrix metalloproteinases involved in matrix degradation and plaque instability [55]. It is thus possible that, in our study, the increased percentage of M2 monocytes would contribute to the progression of the lesions. Second, the M2 polarization is anti-atherogenic, thanks to the M2 monocyte-macrophage anti-inflammatory actions and their role in apoptotic cell efferocytosis. Studies like that of Sharma et al. showed that M2 macrophages can be protective against atherogenesis, at least in mice. They observed that *ApoE^−/−^* mice deficient in *Klf4*, a transcription factor responsible for M2 polarization, develop more inflammation and atherosclerotic lesions than *ApoE^−/−^* mice [56]. Hence, in our study, the percentage of M2 monocytes would increase as an attempt to limit the development of the progressing atherosclerotic lesions.

### Limitations

Although we have obtained significant results, our population is small and monocentric. A larger population and in other dialysis centers should be considered. This is why we mention “preliminary study” in the title.

The age of our control population is significantly lower than the population of dialysis patients. This could have an impact on our results. Costantini et al., published an article in 2018 [57] showing the age effect on the M1/M2 phenotype. The authors observed that the blood M2 monocytes were reduced in healthy subjects older than 65 years old. Our results show the opposite to Costantini’s data, which indicates that in our observations the M2 increase is probably due to the dialyzed status. In addition, the correlation between Cl-Tyr/Tyr ratio and the percentage of M2 monocytes (Figure 5) is a second argument in favor of the dialysis’ effect.

There is also another point to emphasize: 11 of the 27 patients had lipid-lowering treatment. A previous work, Fu et al. (2019), showed that statins, in vitro, exerted an anti-inflammatory action that could affect monocyte/macrophage polarization [58]. Our population is too small to observe any effect of treatment with lipid-lowering drugs on the phenotype of circulating monocytes. A larger cohort should make it possible to study an effect if it is relevant in vivo.

Another question also arises, the kinetics of these subpopulations of monocytes. In our study, blood samples were taken just before the induction of dialysis. We do not know what the evolution of these populations is in the hours and days that follow before the next dialysis session.

Unfortunately, at this point we cannot answer these questions in this work. These are elements that it should be interesting to explore in future works.

## 5. Conclusions

These results suggest that the oxidative stress and inflammation, amplified in HD patients, induce the production of M-CSF, which is itself responsible for enhancing the M2 polarization.

Further analyses are required in order to assess whether these circulating M2 monocytes favor renal dysfunction and accelerated atherogenesis in hemodialysis patients or whether they represent an attempt to limit renal tissue damage and atherogenesis that ultimately fails, probably due to the frequency of the dialyses.

## Figures and Tables

**Figure 1 biomedicines-09-00084-f001:**
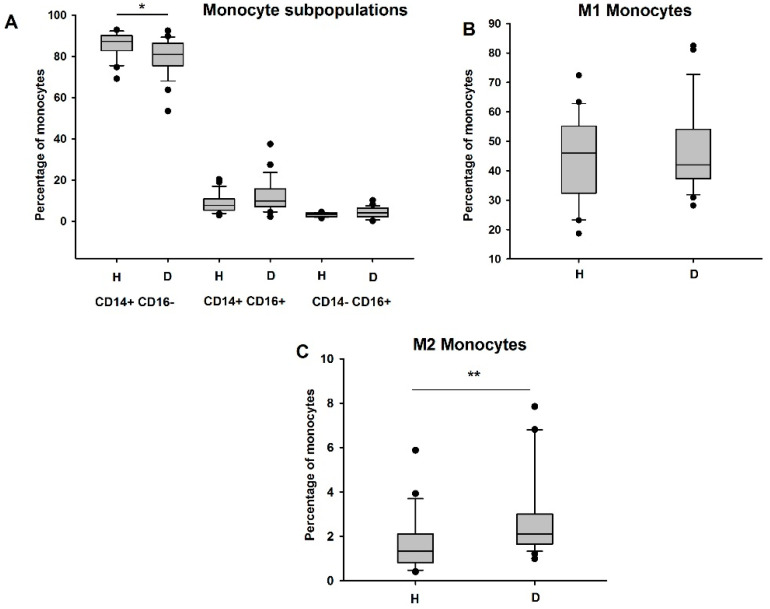
Effects of hemodialysis on monocyte phenotype and polarization. M1 and M2 monocytes in the blood of HD patients (*n* = 27) and healthy subjects (*n* = 23) were analyzed by flow cytometry. (**A**) Quantification of classical, intermediate, and non-classical monocytes. Quantification of M1 monocytes assessed by the positive expression of CD86 and CCR2 (**B**) and of M2 monocytes assessed by the positive expression of CD206, CXCR3, and CD163 (**C**). H: healthy, D: dialyzed patients. * *p* < 0.05; ** *p* < 0.01.

**Figure 2 biomedicines-09-00084-f002:**
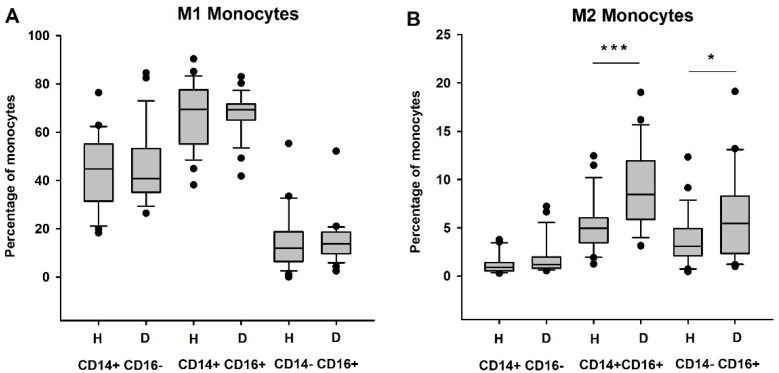
Effects of hemodialysis on monocyte phenotype and polarization. Quantification of M1 (**A**) and M2 (**B**) monocytes in classical, intermediate, and non-classical monocyte subsets. H: healthy, D: dialyzed patients. * *p* < 0.05; *** *p* < 0.001.

**Figure 3 biomedicines-09-00084-f003:**
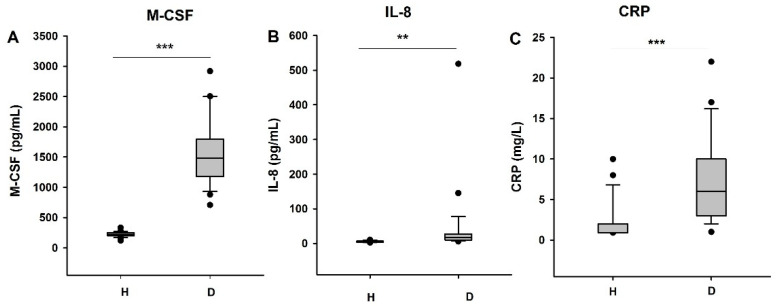
Comparison of the presence of inflammatory factors. (**A**) Quantification of the concentration of M-CSF (pg/mL), (**B**) IL-8 (pg/mL), and (**C**) CRP (mg/L) in the plasma of HD patients (*n* = 27) and healthy volunteers (*n* = 23). H: healthy, D: dialyzed patients. ** *p* < 0.01, *** *p* < 0.001.

**Figure 4 biomedicines-09-00084-f004:**
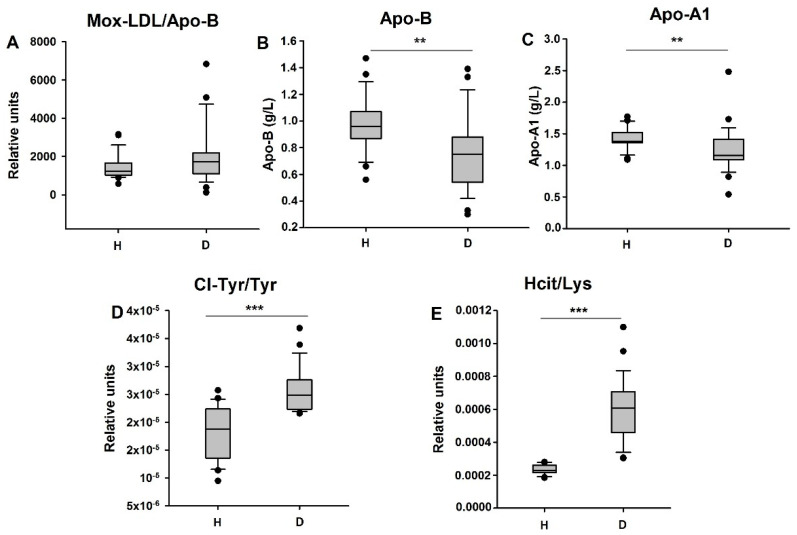
Comparison of the presence of myeloperoxidase-oxidized LDLs and their oxidation products in hemodialyzed patients and in controls. (**A**) Quantification of the plasma concentration of Mox-LDLs (ratio of Mox-LDLs/Apo-B), (**B**) Apo-B (g/L), (**C**) Apo-AI (g/L), and (**D**) chloro-tyrosine (ratio of chloro-tyrosine/tyrosine) in HD patients and healthy subjects. (**E**) Quantification of the plasma concentration of homocitrulline (ratio of homocitrulline/lysine). ** *p* < 0.01; *** *p* < 0.001.

**Figure 5 biomedicines-09-00084-f005:**
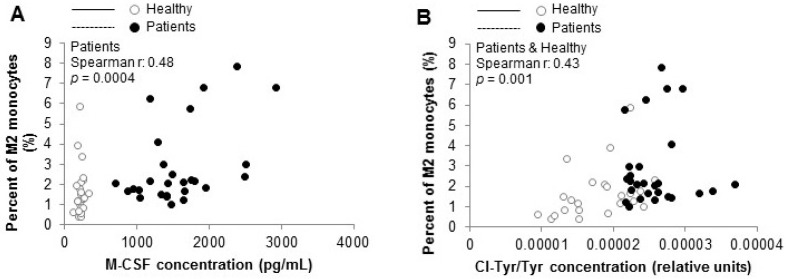
Correlations between the percentage of M2 monocytes and plasma inflammatory and oxidative stress biomarkers in hemodialyzed patients and controls. Correlation between the percentage of circulating M2 monocytes and the plasma concentration of M-CSF (**A**) and between the percentage of circulating M2 monocytes and the plasma concentration of Cl-Tyr/Tyr (**B**) in HD patients (*n* = 27) and healthy subjects (*n* = 23).

## Data Availability

The datasets used during the current study are available from the corresponding author on reasonable request.

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
