# Peer review of "M2 Monocyte Polarization in Dialyzed Patients Is Associated with Increased Levels of M-CSF and Myeloperoxidase-Associated Oxidative Stress: Preliminary Results"

_biomedicines, 2021, doi:10.3390/biomedicines9010084_

Round 1
Reviewer 1 Report
This article deals with the well-known increased risk for hemodialysis patients to develop accelerated atherosclerosis, becoming the main medium term morbidity cause among them. Authors analyse monocyte polarization towards M1 (pro-inflammatory) or M2 (anti-inflammatory) phenotypes, as well as inflammatory and oxidation markers in plasma. An increased concentration of inflammatory cytokines (M-CSF, CRP, IL-8) and myeloperoxidase-dependent oxidative stress plasmatic markers (Mox-LDL, Cl-Tyr, Hcit), as well as a positive correlation seems to exist between these parameters and the amount of M2 polarized circulating monocytes. These observations are interesting and would provide a supporting inflammatory mediators and oxidative mechanisms behind the accelerated atherosclerosis development and, therefore, potential therapeutic targets.
As minor points to be addressed by the authors are:
- H and D labels are used in the Figure 1 and Figure 2. What do they stand for? I suppose that H is Healthy and D is Diseased, but they should be defined in the legends of the figures.
- In Figure 1A there seems to be a possible data mistake: an outlier point in the CD14-/CD16+ monocyte subpopulation represents approximately 100% of total monocyte population for H subjects, while the other points both in H and D subjects are below 5%. If this would be correct, one should also expect an outlier point around 0% for the CD14+/CD16- subpopulation in H subjects.
- Figure 2 could be rearranged to get a horizontal orientation that would make it easiest to read.
- Regarding the flow cytometry monocyte characterization, the authors use isotype controls although for this kind of multiparametric analyses FMO (fluorescence minus one) controls are usually recommended or even needed. Do the authors provide some justification for this technical choice?
Additionally, some points should be clarified by the authors:
- The size of Control and Hemodialysis (HD) patients is small. This limitation should be admitted by the authors as their conclusions may be affected by this fact.
- It is particularly striking the difference in mean ages reported for the study groups: 71+/-14 for HD patients vs. 51+/-28 for Controls. Is this difference statistically significant? Authors could consider the effect of this difference on their conclusions.
- ApoB and ApoA1 significantly lowered in HD patients. Do they were on hypolipemic treatment? Authors should include this information and consider that some hypolipidemic drugs (e.g. statins) do exert anti-inflammatory action and possibly an effect on monocyte/macrophage polarization (Fu, H., Alabdullah, M., Großmann, J. et al.The differential statin effect on cytokine production of monocytes or macrophages is mediated by differential geranylgeranylation-dependent Rac1 activation. Cell Death Dis 10, 880 (2019). https://doi.org/10.1038/s41419-019-2109-9).
- A significant decreased amount of CD14+/CD16- monocytes (classical) in HD patients is described but this fact is not further discussed regarding the features of the studied subjects, nor considered as having a possible relationship with the finding that increased M2 polarization is present in intermediate (CD14+/CD16+) and non-classical (CD14-/CD16+). Have the authors thought about the implications of this for their conclusions?
Author Response
Reviewer1:
This article deals with the well-known increased risk for hemodialysis patients to develop accelerated atherosclerosis, becoming the main medium term morbidity cause among them. Authors analyse monocyte polarization towards M1 (pro-inflammatory) or M2 (anti-inflammatory) phenotypes, as well as inflammatory and oxidation markers in plasma. An increased concentration of inflammatory cytokines (M-CSF, CRP, IL-8) and myeloperoxidase-dependent oxidative stress plasmatic markers (Mox-LDL, Cl-Tyr, Hcit), as well as a positive correlation seems to exist between these parameters and the amount of M2 polarized circulating monocytes. These observations are interesting and would provide a supporting inflammatory mediators and oxidative mechanisms behind the accelerated atherosclerosis development and, therefore, potential therapeutic targets.
As minor points to be addressed by the authors are:
- H and D labels are used in the Figure 1 and Figure 2. What do they stand for? I suppose that H is Healthy and D is Diseased, but they should be defined in the legends of the figures.
We have added in each legend of the figures the definition of "H" and "D".
- In Figure 1A there seems to be a possible data mistake: an outlier point in the CD14-/CD16+ monocyte subpopulation represents approximately 100% of total monocyte population for H subjects, while the other points both in H and D subjects are below 5%. If this would be correct, one should also expect an outlier point around 0% for the CD14+/CD16- subpopulation in H subjects.
Indeed, there was an error in the figures. They have been corrected and we have subsidized the figures to increase the readability. Now there are five figures.
- Figure 2 could be rearranged to get a horizontal orientation that would make it easiest to read.
Done (cf: point 2)
- Regarding the flow cytometry monocyte characterization, the authors use isotype controls although for this kind of multiparametric analyses FMO (fluorescence minus one) controls are usually recommended or even needed. Do the authors provide some justification for this technical choice?
We share the reviewer's opinion regarding the using FMO (fluorescence minus one). it is a technique that has now been proven in flow cytometry. We could have done this to determine the three main populations CD14+/CD16-, CD14+/CD16+ and CD14-/CD16+. However, FMO is very valuable when there is sufficient cell density. As far as we are concerned, we had to look in a trail of cells, especially for CD14+/CD16- and CD14-/CD16+ of subpopulations M1 and M2. In this situation it is difficult to use the FMO. This is where the use of isotypic antibodies comes in handy.
For the sake of consistency for the work, we used the same technique for all conditions.
In addition, it allowed us to find the values that we had obtained in a previous work (k Zouaoui Boudjeltia et al. Monocyte-platelet complexes on CD14/CD16 monocytes subsets: relationship with ApoAI levels. Cardiovascular Pathology 2008:17;285-288).
Additionally, some points should be clarified by the authors:
- The size of Control and Hemodialysis (HD) patients is small. This limitation should be admitted by the authors as their conclusions may be affected by this fact.
- It is particularly striking the difference in mean ages reported for the study groups: 71+/-14 for HD patients vs. 51+/-28 for Controls. Is this difference statistically significant? Authors could consider the effect of this difference on their conclusions.
We combine our responses for the points 1 and 2.
Indeed, we observe a statistical difference between the Controls 55[52-58] years and the HD 71[63-83] years, p<0.001. We presented results in non-parametric data. This point has been added in the 3.1 rubric.
Previously, Costantini A et al, published an article (Age-related M1/M2 phenotype changes in circulating monocytes from healthy/unhealthy individuals. Aging 2018.(10);6:1268-1280) where M1/M2 monocytes were determined in a population younger than 65 years old and in a second one older than 65 years old. Authors observed that the M2 monocytes were reduced in subjects older than 65 years old. In adipose tissues, the same observation is reported. There is an increase in the M1/M2 macrophages ratio (Zeyda et al.2007; Chinetti-Gbaguidi & Staels, 2011)
In bloodstream, these data are opposite to our observations indicating that in our results the M2 increase is probably due to the dialyzed status.
This comment has been added in the Limitations paragraph in the discussion.
- ApoB and ApoA1 significantly lowered in HD patients. Do they were on hypolipemic treatment? Authors should include this information and consider that some hypolipidemic drugs (e.g. statins) do exert anti-inflammatory action and possibly an effect on monocyte/macrophage polarization (Fu, H., Alabdullah, M., Großmann, J. et al.The differential statin effect on cytokine production of monocytes or macrophages is mediated by differential geranylgeranylation-dependent Rac1 activation. Cell Death Dis10, 880 (2019). https://doi.org/10.1038/s41419-019-2109-9).
We added comments about statin treatment in our dialyzed patients (11 patients treated on 27) in the Limitations paragraph. The reference proposed by reviewer has been added.
- A significant decreased amount of CD14+/CD16- monocytes (classical) in HD patients is described but this fact is not further discussed regarding the features of the studied subjects, nor considered as having a possible relationship with the finding that increased M2 polarization is present in intermediate (CD14+/CD16+) and non-classical (CD14-/CD16+). Have the authors thought about the implications of this for their conclusions?
Thanks to the reviewer for bringing this to our attention.
What we see is that the decrease observed in dialysis patients on the CD14 + / CD16- (p = 0.02) population is associated with a slight increase in the CD14 + / CD16 + (p = 0.09) and CD14- / CD16- populations (p = 0.1), without this being statistically significant (Fig1A). This seems to result in an increase in the M2 subpopulation in CD14 + / CD16- and CD14- / CD16 + (Fig 2B).
This comment has been added in the discussion line 302.
Reviewer 2 Report
Macrophages are key players in renal injury, inflammation, and fibrosis. Renal injury can activate molecular pathways that stimulate the differentiation of macrophages into a pro-inflammatory (M1) or anti-inflammatory (M2) phenotypes induced by cytokines, which contribute to tissue regeneration and repair. The current manuscript by Pireaux et al. demonstrates the role of inflammatory cytokines to M-CSF in HD patients leading to enhancing the anti-inflammatory M2 polarization. Overall, the manuscript was mostly well written with a good study design. The reviewer has the following comments:
- As the M2 phenotypes contribute to tissue regeneration and repair, the manuscript clearly leads to one obvious question, what are the follow-up levels of these macrophages? It'll be an interesting and helpful addition to this study.
- The authors should include data on potential indicators of kidney function and a possible recovery in patients, including serum creatinine, interdialytic weight, GFR, etc.
- M2 macrophages can densely express VEGF, which is associated with improved renal hemodynamics. It will be interesting if the authors can demonstrate the VEGF levels in controls and dialyzed subjects.
- The quality of the graphical representation of data should be improved. Additionally, there are some missing lower limit lines in the graphs for instance in Fig. 1C and fig1D.
Author Response
Macrophages are key players in renal injury, inflammation, and fibrosis. Renal injury can activate molecular pathways that stimulate the differentiation of macrophages into a pro-inflammatory (M1) or anti-inflammatory (M2) phenotypes induced by cytokines, which contribute to tissue regeneration and repair. The current manuscript by Pireaux et al. demonstrates the role of inflammatory cytokines to M-CSF in HD patients leading to enhancing the anti-inflammatory M2 polarization. Overall, the manuscript was mostly well written with a good study design. The reviewer has the following comments:
- As the M2 phenotypes contribute to tissue regeneration and repair, the manuscript clearly leads to one obvious question, what are the follow-up levels of these macrophages? It'll be an interesting and helpful addition to this study.
This point raised by the reviewer is very interesting.
We also ask ourselves the question of the kinetics of these subpopulations of monocytes. In our study, blood samples were taken just before the induction of dialysis. We do not know what the evolution of these populations is in the hours and days that follow before the next dialysis session.
Unfortunately for this point we cannot answer it in this work. These are elements that we would like to explore in our future works.
A comment was made in the discussion of the article, in the Limitations paragraph.
- The authors should include data on potential indicators of kidney function and a possible recovery in patients, including serum creatinine, interdialytic weight, GFR, etc.
These results regarding patients have been added to the results section (Results 3.1) healthy volunteers and dialyzed patients (Median [25%-75]).
- M2 macrophages can densely express VEGF, which is associated with improved renal hemodynamics. It will be interesting if the authors can demonstrate the VEGF levels in controls and dialyzed subjects.
Thanks to the reviewer for his idea about VEGF, it's an extremely interesting path.
However, we are already on another path, we have been working for several years on Hemeoxygenase-1 (HO-1) and macrophages.
-Antiox Redox Signal 2010;13(10):1491-1502
-Cell Sign 2012;24(1):199-213
-Mediators Inflamm 2016;8249476
During dialysis there is always a low level of hemolysis that induce the release of hemoglobin well known to increase the expression of HO-1.
In 2009, N.Weis (Molecular biology of the cell 20,1280-1288) and colleagues first described the involvement of HO-1 in macrophage polarization toward a M2 phenotype. HO-1 inhibition/deletion is associated with a lack of M2 macrophages and a simultaneous excess of M1 inflammatory macrophages in mice (Devay L et al. Molecular therapy: the journal of the American Society of Gene Therapy 2009;17:65-72). Thus it seems that HO-1 influences a switch to M2 phenotype, which may explain, at least in part, its anti-inflammatory properties. However, the molecular mechanism of macrophage polarization mediated by HO-1 remain unclear (Naito Y et al. Archives of Biochemistry and Biophysics 2014;564:83-88).
We would like to investigate HO-1 in our next work in dialysis.
- The quality of the graphical representation of data should be improved. Additionally, there are some missing lower limit lines in the graphs for instance in Fig. 1C and fig1D
To improve the readability and quality of the graphics, we have subdivided them. Now there are five figures.
Round 2
Reviewer 2 Report
The reviewer is satisfied with the authors' response. However, the figures have to be consistent for quality and the artwork still needs revision.
Author Response
In agreement with the editorial office we resubmit the new clean version of the manuscript.